# Icariin Alleviates *Escherichia coli* Lipopolysaccharide-Mediated Endometritis in Mice by Inhibiting Inflammation and Oxidative Stress

**DOI:** 10.3390/ijms231810219

**Published:** 2022-09-06

**Authors:** Aftab Shaukat, Irfan Shaukat, Shahid Ali Rajput, Rizwan Shukat, Sana Hanif, Shucheng Huang, Muhammad Tahir Aleem, Kun Li, Qiao Li, Chao Chen, Xinxin Zhang, Haimiao Lv, Zhiqiu Yao, Liguo Yang

**Affiliations:** 1National Center for International Research on Animal Genetics, Breeding and Reproduction (NCIRAGBR), Huazhong Agricultural University, Wuhan 430070, China; 2Department of Biochemistry, University of Narowal, Narowal 51600, Pakistan; 3Department of Animal Feed and Production, Faculty of Veterinary and Animal Sciences, Muhammad Nawaz Shareef University of Agriculture, Multan 66000, Pakistan; 4Faculty of Food, Nutrition & Home Sciences, University of Agriculture, Faisalabad 38000, Pakistan; 5Hubei Key Laboratory of Theory and Application of Advanced Materials Mechanics, Wuhan University of Technology, Wuhan 430070, China; 6College of Veterinary Medicine, Henan Agricultural University, Zhengzhou 450002, China; 7College of Veterinary Medicine, Nanjing Agricultural University, Nanjing 210095, China

**Keywords:** endometritis, icariin, lipopolysaccharide, TLR4, NF-κB and Nrf2 pathway

## Abstract

Icariin (ICA) is a naturally occurring phytochemical agent primarily extracted from *Epimedium Brevicornum* Maxim (Family Berberidaceae) with a broad spectrum of bioactivities. Endometritis is a uterine disease that causes enormous losses in the dairy industry worldwide. In this study, anti-inflammatory and anti-oxidant properties of ICA were investigated against lipopolysaccharide (LPS)-induced endometritis in mice to investigate possible underlying molecular mechanisms. Sixty heathy female Kunming mice were randomly assigned to four groups (*n* = 15), namely control, LPS, LPS + ICA, and ICA groups. The endometritis was induced by intrauterine infusion of 50 µL of LPS (1 mg/mL). After 24 h of onset of LPS-induced endometritis, ICA groups were injected thrice by ICA intraperitoneally six hours apart. Histopathological examination, enzyme linked immunosorbent assay (ELISA), real time quantitative polymerase chain reaction (RT-qPCR), western blotting, and immunohistochemistry were used in this study. Histological alterations revealed that ICA markedly mitigated uterine tissue injury caused by LPS. The results showed that the ICA inhibited the production of pro-inflammatory cytokines (IL-1ß, IL-6, and TNF-α) and boosted the production of anti-inflammatory cytokines (IL-10). Additionally, ICA modulated the expression of malondialdehyde (MDA), reactive oxygen species (ROS), superoxide dismutase 1 (SOD1), catalase (CAT), and glutathione peroxidase 1 (Gpx1) induced by LPS. The administration of ICA significantly (*p* < 0.05) improved the mRNA and protein expression of Toll-like receptor (TLR) 4. The western blotting and ELISA finding revealed that the ICA repressed LPS-triggered NF-κB pathway activation. Moreover, ICA improved the antioxidant defense system via activation of the Nrf2 pathway. The results revealed that ICA up-regulated the mRNA and protein expression of Nuclear erythroid-2-related factor (Nrf2), NAD(P)H: quinone oxidoreductase 1 (NQO1), heme oxygenase-1 (HO-1), and glutamate-cysteine ligase catalytic subunit (GCLC) under LPS exposure. Conclusively, our findings strongly suggested that ICA protects endometritis caused by LPS by suppressing TLR4-associated NF-κB and Nrf2 pathways. Altogether, these innovative findings may pave the way for future studies into the therapeutic application of ICA to protect humans and animals against endometritis.

## 1. Introduction

Endometritis is an inflammation of the uterine endometrium, causing substantial economic losses to the dairy industry [1]. The disease is characterized by foul, odorous, pyogenic uterine secretion, depression, fever, and dehydration [2]. Various microbial pathogens are involved in the pathogenesis of endometritis, including *Escherichia coli* (*E. coli*), which is Gram-negative bacteria [3,4]. *E. coli* causes subclinical and clinical endometritis [5]. Presently, antibiotics have been used to treat uterine bacterial diseases. Misuse of excessive antibiotics in the dairy industry may lead to antibiotic resistance, and antimicrobial residues in milk and meat that cause serious food safety issues globally [6]. Therefore, novel preventive and therapeutic strategies for treating endometritis in the dairy industry are urgently needed. To minimize the costs, mouse models have already been used to study the underlying mechanism of bovine endometritis [1,5,7]. In the present study, we utilized a mouse model to investigate endometritis.

The endometrium is the first layer of protection in the uterus that have a vital role in the pathophysiology of invading organism-induced endometritis [8,9,10]. Lipopolysaccharide (LPS) is an endotoxin obtained from Gram-negative bacteria’s cell walls. LPS also serves as an etiological agent causing inflammation in the host [11]. The LPS elicits the activation of Toll-like receptor (TLR) 4. After being recognized by TLR4, several signaling pathways were activated resulting in the overproduction of pro-inflammatory markers and oxidative stress [10,12].

When cells are under oxidative stress caused by inflammatory responses, reactive oxygen species (ROS) contents in the tissues can rise. This can have numerous detrimental effects on the tissues [13]. ROS has previously been linked to the development of various inflammatory disorders, including endometritis [14]. Anti-oxidative enzymes (CAT, SOD1, and Gpx1) and Nrf2 are believed to be crucial in ROS-mediated pathologic conditions in mice [15,16]. In recent research, Shaukat et al., have shown that *S. aureus* triggered oxidative stress via ROS production in mice [15]. A high quantity of ROS inside the murine alveolar cells can activate the NF-κB pathway, which is involved in several pathogenic events, including bacteria-induced inflammation [17]. NF-κB stimulates the secretion of pro-inflammatory cytokines, exacerbating bovine and murine cell and tissue damage [18]. As a result, inhibiting ROS-mediated NF-κB activation can be a useful treatment for inflammatory illnesses such as endometritis.

Recently, numerous Chinese herbal medicines have been extensively used to treat diseases such as mastitis [19], acute lung injury [15,20], and endometritis [5,9]. Icariin (ICA) is a type of natural flavonoid isolated from *Epimedium Brevicornum* Maxim (Figure 1A,B). ICA was utilized as a tonic effect many years ago in China [21]. ICA have a diverse range of pharmacological properties, including immunoregulatory [22], antioxidant [23], antidepressant [24], cardiomyocyte differentiation [25], stimulation of angiogenesis [26], and anti-inflammatory activities [27]. However, the anti-inflammatory and anti-oxidant properties of ICA on endometritis caused by LPS have not yet been elucidated. Here, we used a mouse model of *E. coli*-derived LPS-induced endometritis in this investigation to evaluate whether ICA can attenuate the endometritis. To our knowledge, this is the first study to document the protective role of ICA against LPS-induced endometritis.

## 2. Results

### 2.1. Effect of ICA against LPS-Inflicted Murine Endometritis

There are no inflammatory changes in the control group. It was shown that infusion of LPS consequently leads to severe injury, including infiltration of inflammatory cells, hyperemia, and hemorrhage. However, pathological changes induced by LPS were improved by ICA (Figure 2A). The scoring of the histopathological section was carried out and revealed the histopathological alterations (Figure 2B).

### 2.2. Effect of ICA on W/D Ratio, MPO Activity, and NO Concentration Assay

Edema is a characteristic feature of LPS-induced endometritis. The W/D ratio was detected to assess the extent of the inflammatory uterine edema. Exposure of LPS dramatically increased (*p* < 0.05) the W/D ratio. The increased W/D ratio was significantly (*p* < 0.05) decreased in ICA groups (Figure 2C). The MPO activity estimates the penetration of inflammatory cells at the site of inflammation. The results revealed that the MPO activity was enhanced dramatically (*p* < 0.05) in LPS groups compared to the control group. Upon administration of the ICA, the LPS-induced MPO activity is significantly improved (*p* < 0.05) (Figure 2D).

The NO is an essential indicator of inflammation that increases vascular permeability. Our results have demonstrated that LPS has up-regulated the production of NO as compared to the control group, whereas the LPS-stimulated NO production was significantly down-regulated by the treatment of ICA (*p* < 0.05). (Figure 2E).

### 2.3. Outcome of ICA on Cytokines

The ELISA and qRT-PCR were used to explore the consequence of ICA on LPS-triggered expression of the pro-inflammatory and anti-inflammatory cytokines. The results demonstrated that the ICA administration significantly (*p* < 0.05) repressed the LPS-induced protein and gene over-expression of pro-inflammatory cytokines (TNF-α, IL-1β, and IL6). However, the ICA has noticeably (*p* < 0.05) boosted the concentration and gene expression of anti-inflammatory (IL10) cytokine compared to both the control and LPS groups (Figure 3A,B).

### 2.4. Effect of ICA on LPS-Triggered TLR4-Mediated NF-κB Pathway

TLR4 is an important receptor in the inflammatory response of LPS. As shown in the results of the immunohistochemistry (Figure 4A) as well as the qRT-PCR assay (Figure 4B), the expression of TLR4 markedly increased (*p* < 0.05) in the LPS group. On the other hand, ICA therapy inhibited (*p* < 0.05) the LPS-triggered TLR4 expression.

The protective effect of the ICA during LPS-induced murine endometritis was explored by measuring the protein expression of the NF-κB pathway via ELISA assay and western blotting. As demonstrated in the ELISA assay, the expression of the phosphorylated NF-κB p65, and IκB-α was immensely (*p* < 0.05) increased in the LPS group, which is decreased (*p* < 0.05) upon ICA therapy (Figure 5A,B). The NF-κB pathway protein concentration was measured by western blotting assay to confirm our findings. Interestingly, a similar pattern has been observed as in the results of the ELISA assay (Figure 5C,D).

### 2.5. Effect of ICA on LPS-Triggered Oxidative Stress Markers in Endometritis

SOD1, CAT, and Gpx1 are activated throughout the recovery process from oxidative damage. LPS-induced ROS production and MDA formation are significantly (*p* < 0.05) reduced by ICA therapy, and reduced in CAT, SOD1, and Gpx1 depletion significantly (*p* < 0.05). As displayed in Table 1, these results were reversed noticeably (*p* < 0.05) using ICA. The mRNA expression level of SOD1, CAT, and Gpx1 were markedly (*p* < 0.05) decreased in the LPS group Figure 6A. However, ICA treatment showed a significant up-regulation of these enzymes. These outcomes demonstrated that ICA reduces the oxidative damage caused by LPS in uterine tissue.

### 2.6. Effect of ICA against LPS-Triggered Activation of Nrf2 Pathway

When LPS was administered, the mRNA expression levels of Nrf2 and downstream genes (NQO1, HO-1, and GCLC) were considerably (*p* < 0.05) lower in the LPS group than in the control. However, the mRNA expression level of Nrf2 and its downstream genes in uterine tissue was significantly (*p* < 0.05) up-regulated with ICA treatment (Figure 6B).

Next, LPS affects down-regulation of Nrf2, HO-1, and NQO1 protein expression compared to the control group, which was up-regulated by ICA therapy (Figure 6C,D). These findings suggested that ICA could attenuate LPS-induced oxidative stress via activating the Nrf2 pathway.

## 3. Discussion

Endometritis is the inflammation of the endometrial layer of the uterine wall and has severely slowed the improvement of the dairy industry [6]. However, the antibiotics used in therapeutic measures are effective but lead to resistance in bacteria and food safety problems. Traditional Chinese medicines have been used globally for their prophetic and therapeutic uses in inflammatory diseases. Icariin (ICA) is the primary bioactive component isolated from *E. brevicornum* [21], and has diverse biological activities [28]. Moreover, ICA has been demonstrated to attenuate the production of pro-inflammatory mediators via the inhibition of NF-kB [21], and modulate the redox reaction by the Nrf2 pathway [29]. However, to be best of our knowledge, no prior research studies had explored the effects of ICA in endometritis. LPS originates from the cell walls of Gram-negative bacteria responsible for inflammatory pathogenic response [2,5,9,30]. The LPS-induced mice model of endometritis is a well-established model used to explore the effectiveness of treatment strategies in endometritis [30,31]. In the current study’s findings, the morphologic alteration has been observed in the LPS-administered murine uterus. The changes were attenuated after ICA therapy. Furthermore, the histology results of the uterine tissue sample revealed that the treatment of ICA dramatically decreases the accumulation of inflammatory cells in the pathological injury of uterine tissues, strongly suggesting that ICA has a protective effect on endometritis induced by LPS. Edematous swelling is a characteristic feature of LPS-induced endometritis [3]. The MPO activity is to estimate the infiltration of neutrophils at the site of inflammation. The NO is an important indicator of inflammation that increases vascular permeability. Subsequently, the massive neutrophils and macrophages approach the inflammatory site to boost the inflammation. The results of our study indicate that MPO activity, W/D ratio, and NO concentration were significantly reduced by ICA compared to LPS. The results of our study are consistent with those of previous research [27].

Inflammatory reactions are beneficial in response to harmful stimuli, such as microbial pathogens, irritants, and apoptotic cells [32]. At the same time, inflammation can be life-threatening and may severely damage the body tissues in chronic or severe acute inflammation [33]. The enhanced production of pro-inflammatory markers in LPS-induced endometritis causes severe uterine injury [34]. The IL-1β and TNF-α have been considered as the primary pro-inflammatory cytokines produced by several types of immune cells, such as activated monocytes, macrophages [35], and epithelial cells [36], which could stimulate the secreting of other inflammatory mediators. Pro-inflammatory cytokines stimulate the activation of cell adhesion molecules, which enhances the migration and adhesion of leukocytes to the site of inflammation [37]. Therefore, we hypothesized that attenuation of pro-inflammatory cytokines might improve the endometritis outcomes. We measured the expression level of the pro-inflammatory cytokines to explore whether ICA has an inhibitory effect on producing these inflammatory mediators. Additionally, the expression of anti-inflammatory cytokine (IL-10) may also be increased by some potent anti-inflammatory drugs [7,19]. Intriguingly, treatment of ICA significantly repressed the expression of IL-1β, IL-6, and TNF-α as well as boosted the production of IL-10. The findings of current research are in-line with the previous reports [27,38]. Hence, we concluded that ICA might express its protective effect by attenuation of the production of IL-1β, IL-6, and TNF-α as well as boosting the production of IL-10 in LPS-induced endometritis.

NF-κB is a crucial nuclear transcription factor that plays an essential role in regulating the immune response in various inflammatory diseases. NF-κB has been known to regulate the transcription of several genes that control the production of chemokines and cytokines in inflammatory diseases [39]. Generally, under normal physiological conditions, NF-κB is found as an inactive form bounded to its inhibitor IκBα in the cytoplasm; upon inflammatory stimulation by LPS, IκB-α is degraded and phosphorylated [40]. The NF-κB p65 is dissociated from the IκBα unit and translocated to the nucleus as an active form where it regulates the activation of inflammation and immune response-related genes such as TNF-α and IL-1β [41]. Next, we quantify the expression levels of p-p65 and p-IκBα. Our result revealed that ICA treatment dramatically decreases the level of phosphorylation of NF-κb-p65 and IκBα induced by LPS, suggesting that ICA have an inhibitory effect on LPS-induced activation of NF-κB. TLR4 acts as a crucial triggering receptor in LPS-challenged activation [34]. It is well reported that LPS triggers immunological disorders via TLR4 and the downstream NF-κB pathway [3,5,9,30,35]. Indeed, previous studies have verified that LPS induces the production of cytokines by triggering the activation of the TLR4 pathway [2,17]. As expected, LPS treatment significantly up-regulated the expression of TLR4, which was down-regulated by ICA treatment. Therefore, it is hypothesized that ICA therapy inhibited NF-κB activation by reducing the TLR4 expression. Additionally, we also validated the function of TLR4 in LPS-induced endometritis immuniohistochemistry. These findings advocated the anti-inflammatory action of ICA via TLR4-mediated NF-κB signaling pathway.

Redox homeostasis is necessary for the normal functioning of the body. It has been described that LPS increases ROS generation and accumulation of MDA, meanwhile decreasing antioxidant defense [17,42]. MDA accumulation is a primary indication of oxidative stress and lipid peroxidation [15]. The results of previous studies [37,43], have shown that LPS could cause oxidative damage. Antioxidant defense mechanisms reduced ROS, hence reducing ROS-induced damage. In the current study, we observed that the accumulation of MDA contents and ROS in the uterine tissue was increased due to LPS exposure, and the ICA therapy attenuated the MDA concentration and ROS. These findings depicted that ICA inhibited the LPS-induced ROS-dependent lipid peroxidation in the murine uterus.

Antioxidant enzymes are abundant in cells e.g., CAT, SOD, and Gpx [44]. SOD converts superoxides to hydrogen peroxide and oxygen. CAT catalyzes hydrogen peroxide into atmospheric oxygen and water. Gpx metabolize hydrogen peroxide into nontoxic components [45]. ICA attenuates oxidative stress markers (SOD, CAT, Gpx, and glutathione reductase) in rats [29]. In this study, ICA drastically boosted the production of SOD, CAT, and Gpx. These findings revealed that ICA has a protective effect in LPS-induced endometritis which is due to antioxidant enzyme activities.

Nrf2 is regarded as the “master regulator” and has been shown to defend against oxidative stress in various diseases including endometritis [12,45,46]. The Nrf2 pathway has an antioxidative impact, which helps to reduce LPS-induced oxidative damage [12]. The activation of the Nrf2 pathway protects the cell from oxidative damage [30]. When Nrf2 is activated, various downstream genes, such as HO-1, NQO1, and GCLC, which are critical components of the endogenous redox system, may be activated as well. In the case of oxidative damage, these proteins have cytoprotective properties [47,48]. In the previous studies, it has been described that ICA improves neuro-inflammation, synovitis and suppress oxidative stress by activation of the Nrf2 pathway [49,50]. However, the comprehensive mechanism through which ICA alleviates oxidative stress via Nrf2 pathway in LPS-induced endometritis remains to be investigated. Therefore, in this experiment, the therapeutic effect of ICA in LPS-induced oxidative damage via regulating the Nrf2 pathway in the uterine tissue of mice has been explored. Interestingly, it was observed that LPS downregulate Nrf2 and its downstream genes (HO-1, NQO1, and GCLC) in uterine tissue. Our findings are consistent with previous studies that suggest that ICA can reduce oxidative stress by activating the Nrf2 pathway [49,50].

## 4. Materials and Methods

### 4.1. Ethical Statement

The study was approved by the animal ethical board of Huazhong Agricultural university (HZAUMO-2015-12), Wuhan, China.

### 4.2. Reagents

ICA was obtained from Royal pharm, Hanzhou, China. *E. coli* strain [O55:B5]-derived LPS was acquired from Sigma (St. Louis, MO, USA). ELISA kits of superoxide dismutase (SOD), malondialdehyde (MDA), catalase (CAT), glutathione peroxidase (Gpx)1, myeloperoxidase (MPO), nitric oxide (NO), tumor necrosis factor (TNF)-α, interleukin (IL)-1β, IL-6, and IL-10 were obtained from Nanjing Jiancheng bioengineering institute (Nanjing, China). The primary and secondary antibodies (TLR4, NF-κB pathway proteins and β-actin) were obtained from cell signaling Technology (CST, Beverly, MA, USA).

### 4.3. HPLC Analysis of ICA

The HPLC was carried out to assess the purity of ICA by using the EChrom2000 DAD data system according to our published method [15]. Briefly, the chromatography was performed via Hyper ODS (250 × 4.6 mm, 5 µm Dikma, Lake Forest, CA, USA). The acetonitrile in water (2:98) was utilized for elution purposes. The flow speed was 1 mL/minute, and the detection wavelength was 295 nm. The purity of ICA was 98% (Figure 1C).

### 4.4. Animals

Adult female Kunming mice (60 females, weighing 30 ± 2 g and 8–12 weeks of age) were purchased from the Animal Centre of Wuhan University (Wuhan, China). The animals had ad libitum access to a standard diet and fresh drinking water throughout the experimental period. The mice were housed under standard conditions at 24 ± 1 °C and 65% humidity and kept on twelve hours (h) of light and twelve hours of dark.

### 4.5. Treatment Design

The mice were allocated randomly into four treatment groups (*n* = 15), as mentioned in Table 2. The required concentrations of LPS and ICA were prepared by dissolving them into sterile phosphate-buffered saline (PBS) and ethanol, respectively. The ICA dose-rate selection was based on previous research [29,51]. A mice model of LPS-induced endometritis was developed as described previously [2,5]. The endometritis was induced by intrauterine infusion of 50 µL of LPS (1 mg/mL). After 24 h of onset of LPS-induced endometritis, ICA groups were injected thrice by ICA intraperitoneally six hours apart. After euthanization, the middle part of the uterine horns from every group was collected and stored at −80 °C for further experiments.

### 4.6. Histopathologic Assay of Uterine Tissue

The histopathological assay was carried out to evaluate the pathological lesions in uterine tissue induced by LPS. The uterine tissue from each group was collected, sliced with a microtome to an approximate size of 0.5 cm^3^, and then embedded into paraffin. The uterine tissue was fixed with 10% formalin stained with hematoxylin and eosin (H&E). Histopathological lesions were observed with a light microscope (Olympus, Tokyo, Japan).

### 4.7. Wet to Dry Weight (W/D) Ratio of Uterine Tissue

The uterine wet weight (W) was measured immediately after euthanization by weighing the fresh excised uterine tissue. Next, the dry weight (D) was detected after placing uterine samples in a hot air oven at a temperature of 80 °C for 24 h. The extent of inflammatory edema was calculated as the W/D ratio.

### 4.8. MPO Activity Assay and Determination of NO Concentration

According to the supplier’s instructions, the uterine tissue was homogenized, centrifuged, and the supernatant was collected to determine the MPO activity and NO concentration.

### 4.9. ELISA Analysis

Uterine tissue samples were homogenized, centrifuged, and then supernatants were harvested. The supernatants were used to determine concentrations of TNF-α, IL-1β, IL6, IL10, and NF-κB pathway (total & phospho IκB-α and NF-κB-p65) and according to producer recommendations.

### 4.10. Determination of Oxidative Stress Markers

The MDA concentrations and activity of SOD, CAT, and Gpx were determined according to directions from the manufacturer. The SOD, CAT, and Gpx values are reported in units per milligram (U/mg) and MDA in nanomoles per milligram (nmol/mg) of protein.

### 4.11. qRT-PCR Analysis

The mRNA levels of the relevant genes employed in this investigation were measured using qRT-PCR, as described in our prior study [52]. Briefly, total RNA from tissue samples was extracted with TRIzol reagent (Invitrogen, Carlsbad, CA, USA) according to the manufacturer’s instructions. The concentration and purity of RNA samples were determined using the nucleic acid concentration analyzer NanoDrop 2000 (Thermo Fisher, Waltham, MA, USA) based on the absorbance ratio at 260 and 280 nm. The RNA was then reverse transcribed to cDNA using a reverse transcription kit (Takara, Japan). The cDNA served as a template for subsequent qRT-PCR reactions. The qRT-PCR analysis was executed using a SYBR Green qRT-PCR kit (Roche, Basel, Switzerland) with Light Cycler 96 (Roche, Basel, Switzerland) according to the instructions of the manufacturer. Primers that were used in research are mentioned in Table 3. Relative transcriptional expression of the target genes was normalized to the control group. Quantification was performed using the 2^–ΔΔCt^ formula, and the GAPDH was used as a reference gene [53].

### 4.12. Immunohistochemistry

The comprehensive procedure for fixation, embedding with paraffin, and slicing of tissue was the same as the H&E staining procedure in Section 2.5. Next, tissue sections were deparaffinized by xylene, followed by incubation with 3% H_2_O_2_ at room temperature for 10 min. The blocking was performed with goat serum for half an hour at 37 °C. Consequently, the primary antibody was probed overnight, and a secondary antibody was applied for one hour. After DAB staining, these sections were counterstained with hematoxylin and viewed under a microscope.

### 4.13. Western Blot Analysis

Total protein from uterine tissue samples was extracted using a RIPA lysis buffer. A BCA kit was utilized for the quantification. Equal amounts of protein were separated using 10% SDS-PAGE and then transferred to PVDF membranes. Following our previous procedure, the membranes were blocked with 5% nonfat milk and then treated with primary and secondary antibodies [40]. A chemiluminescence detection system was used to quantify the intensities (ImageQuant LAS 4000 mini, Cytiva, Marlborough, MA, USA).

### 4.14. Statistical Analysis

Data analysis was carried out by Graphpad prism 9.4.0 (San Diego, CA, USA). All the data were displayed as the means ± SEM. One-way ANOVA followed by Dunnett’s multiple comparison tests were used for statistical analysis. A *p*-value of <0.05 was considered statistically significant.

## 5. Conclusions

In summary, our results revealed that the LPS could intensify inflammatory response and oxidative damage in uterine tissue of mice. Conversely, ICA protects from LPS-induced inflammation and oxidative stress through the modulation of the TLR4-mediated NF-κB and Nrf2 signaling pathways. Taken together, our findings demonstrated that NF-κB and Nrf2 pathways were modulated by ICA, which indicated the potential application of ICA against LPS-induced inflammation and oxidative stress in the uterine tissue of mice. Based on our results, ICA may be useful as an anti-inflammatory medicine for treating inflammatory diseases, such as *E. coli*-induced endometritis.

## Figures and Tables

**Figure 1 ijms-23-10219-f001:**
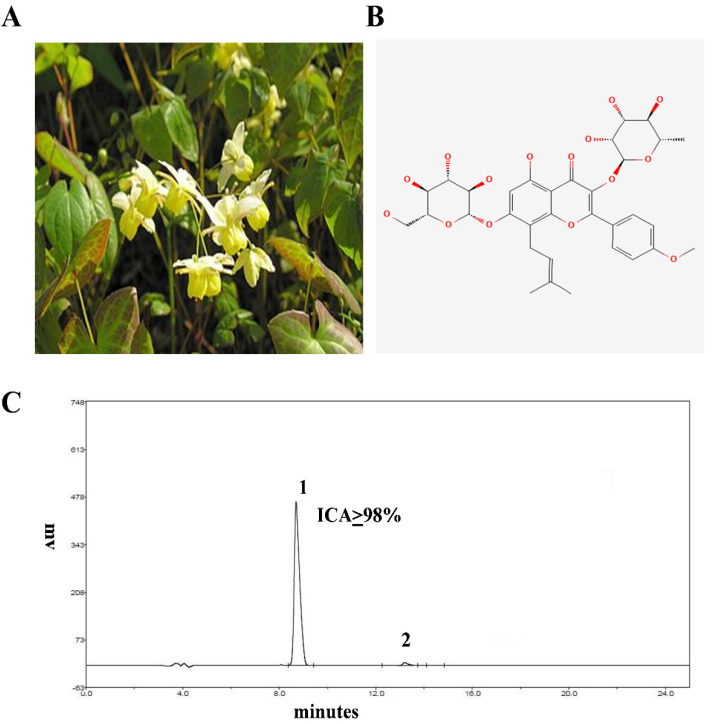
(**A**) *Epimedium Brevicornum*. (**B**) Structure of ICA. (**C**) HPLC chromatogram of ICA (the purity of ICA was more than or equal to 98%).

**Figure 2 ijms-23-10219-f002:**
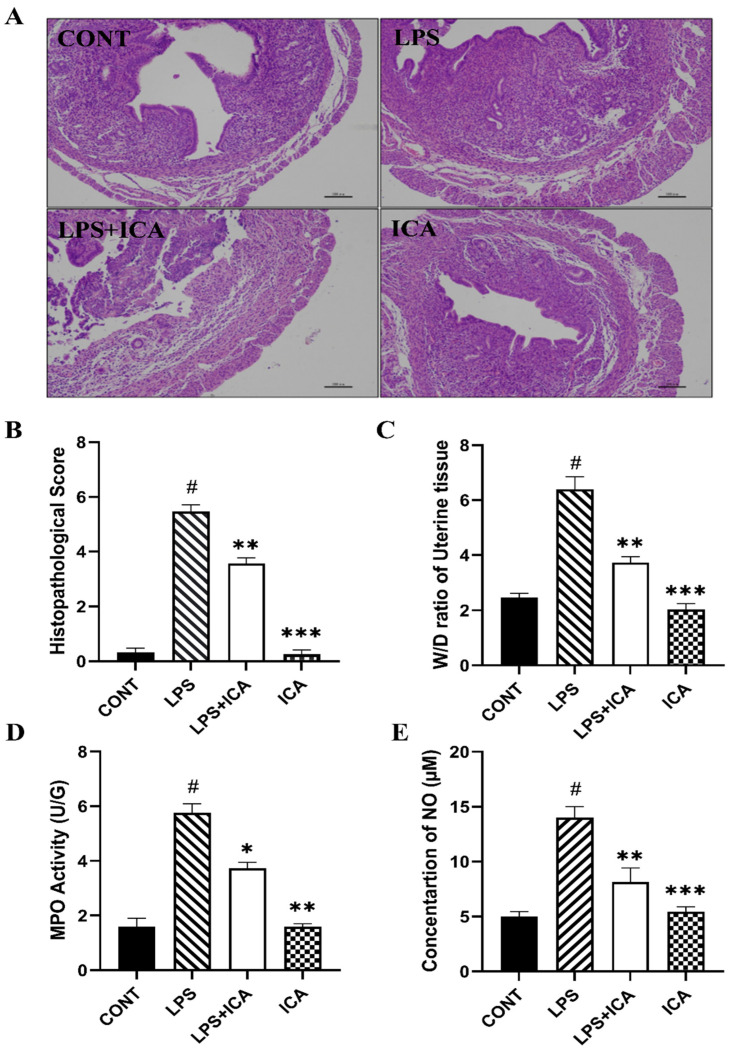
The effect of ICA on LPS-induced uterine injury in mice (**A**) H&E analysis of uterine tissue, Scale bar: 100 μm. (**B**) Histopathological scoring of the murine uterus. (**C**) W/D ratio. (**D**) MPO activity assay and (**E**) NO concentration. The data were statistically presented as means ± SEM. The # *p* < 0.001 between CONT and LPS groups, and * *p* < 0.05, ** *p* < 0.01, *** *p* < 0.001 between LPS and ICA therapy groups. (CONT, LPS, and ICA stand for control, lipopolysaccharide, and Icariin groups [The ICA dose used was 50 mg/kg for uterine tissue]).

**Figure 3 ijms-23-10219-f003:**
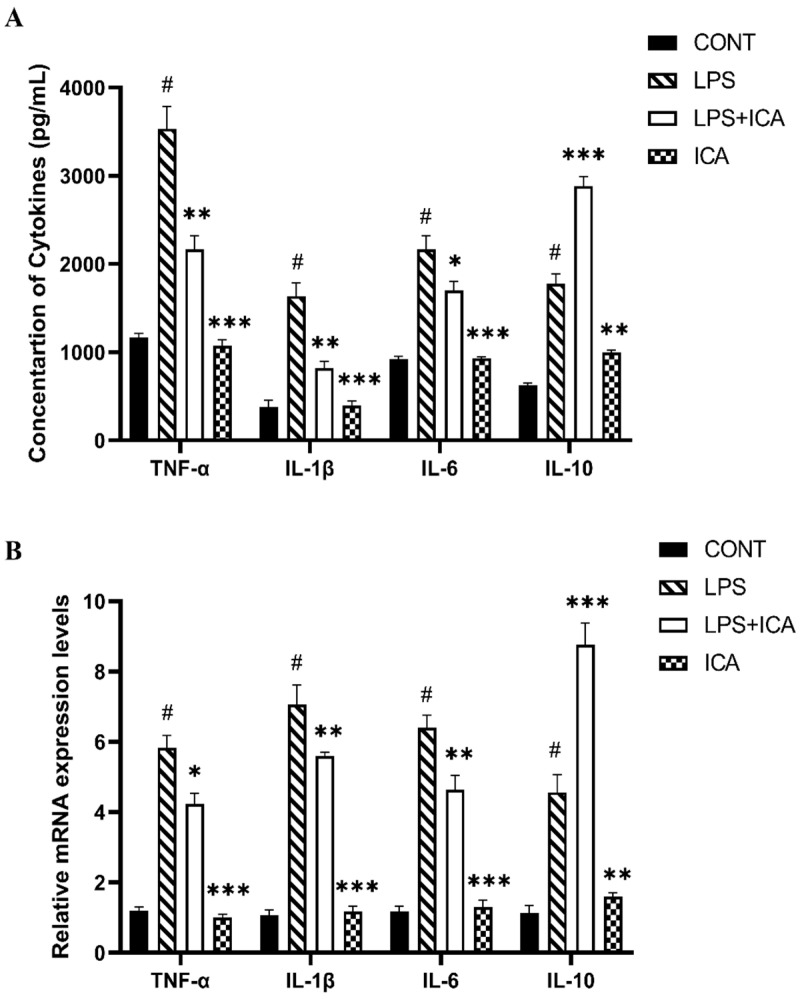
Effect of ICA in LPS-triggered secretion of cytokines. (**A**) Concentration of TNF-α (pg/mL), IL-1β (pg/mL), IL-6 (pg/mL) and IL-10 (pg/mL). (**B**) The relative mRNA expression levels of TNF-α, IL-1β, IL-6, and IL-10. The data were statistically presented as means ± SEM. The # *p* < 0.001 between CONT and LPS groups, and * *p* < 0.05, ** *p* < 0.01, *** *p* < 0.001 between LPS and ICA therapy groups. (CONT, LPS, and ICA stand for control, lipopolysaccharide, and Icariin groups. [The ICA dose used was 50 mg/kg for uterine tissue]).

**Figure 4 ijms-23-10219-f004:**
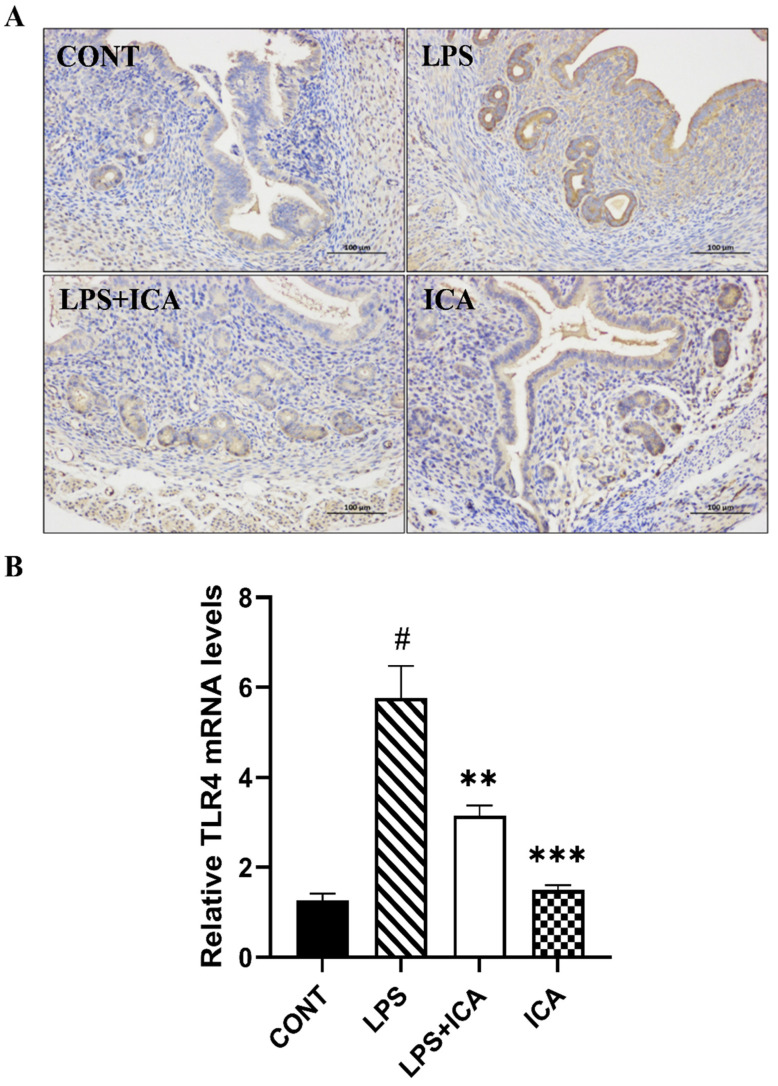
Consequences of ICA on LPS-induced TLR4 expression. (**A**) The immunohistochemical expression of TLR4 protein. (**B**) The relative mRNA expression level of the TLR4 gene. The data were statistically presented as means ± SEM. The # *p* < 0.001 between CONT and LPS groups, and ** *p* < 0.01, *** *p* < 0.001 between LPS and ICA therapy groups. (CONT, LPS and ICA stand for control, lipopolysaccharide and Icariin groups [The ICA dose used was 50 mg/kg for uterine tissue]).

**Figure 5 ijms-23-10219-f005:**
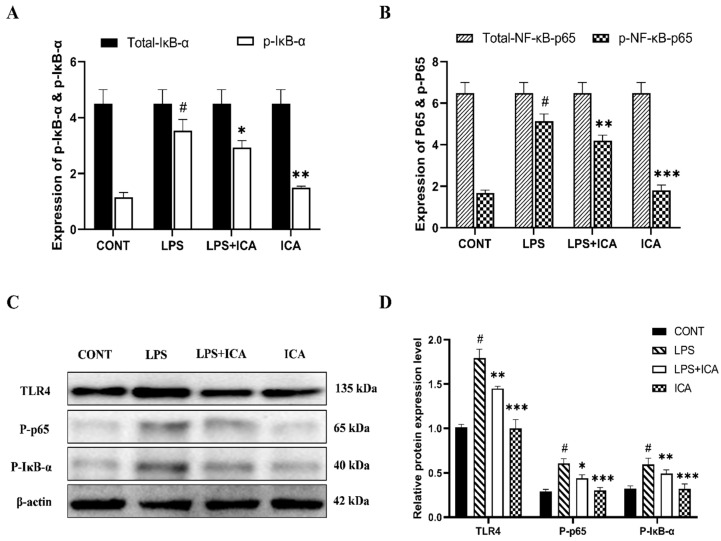
Outcomes of ICA in LPS-induced expression NF-κB signaling pathway. (**A**) The expression levels of the NF-κB-p65 and its phosphorylated (p-NF-κB-p65) form were detected by ELISA. (**B**) The expression levels of total IκBα and its phosphorylated (p-IκBα) form were detected by ELISA. (**C**) The protein expression levels of the TLR4, phosphorylated (p-NF-κB-p65), and phosphorylated (p-IκBα) were detected by western blotting. (**D**) The quantification of all the analyzed proteins. The β-Actin was used as a control. The data were statistically presented as means ± SEM. The # *p* < 0.001 between CONT and LPS groups, and * *p* < 0.05, ** *p* < 0.01, *** *p* < 0.001 between LPS and ICA therapy groups. (CONT, LPS, and ICA stand for control, lipopolysaccharide, and Icariin groups [The ICA dose used was 50 mg/kg for uterine tissue]).

**Figure 6 ijms-23-10219-f006:**
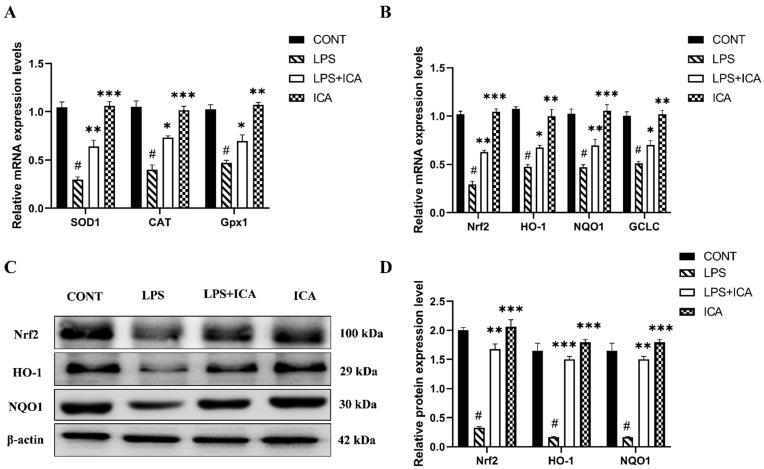
Influence of ICA treatment on LPS-induced oxidative stress-related genes via activation of Nrf2 pathway in murine uterus. (**A**) The relative mRNA expression levels of SOD1, CAT and Gpx1 was determined by RT-qPCR. (**B**) The relative mRNA expression levels of Nrf2, HO-1, NQO1, and GCLC. (**C**) The protein expressions of Nrf2, HO-1, and NQO1 were detected by western blotting. (**D**) The quantification of all the analyzed proteins. The data were statistically presented as means ± SEM. The # *p* < 0.001 between CONT and LPS groups, and * *p* < 0.05, ** *p* < 0.01, *** *p* < 0.001 between LPS and ICA therapy groups. (CONT, LPS, Nrf2, HO-1, NQO1, GCLC, and ICA stand for control, lipopolysaccharide, Nuclear erythroid-2-related factor, heme oxygenase-1, NAD(P)H: quinone oxidoreductase 1, glutamate-cysteine ligase catalytic subunit, and Icariin. [The ICA dose used was 50 mg/kg for uterine tissue]).

**Table 1 ijms-23-10219-t001:** ICA administration mitigated LPS-induced oxidative stress in uterine tissue of mice.

Parameters	CONT	LPS	LPS + ICA	ICA
ROS (florescence/mg protein)	69.23 ± 4.11	276.63 ± 24.64 #	180.19 ± 11.41 **	78.64 ± 6.49 ****
MDA (nmol/mg protein)	2.11 ± 0.31	5.47 ± 0.58 #	4.16 ± 0.33 *	1.99 ± 0.18 ***
SOD (U/mg protein)	41.85 ± 5.44	13.73 ± 2.14 #	27.41 ± 6.97 **	43.44 ± 4.67 ***
CAT (U/mg protein)	69.73 ± 6.45	21.07 ± 5.61 #	47.79 ± 3.31 **	71.34 ± 7.41 ***
Gpx1 (U/mg protein)	199.22 ± 14.42	69.57 ± 5.19 #	127.89 ± 11.18 **	205.01 ± 10.29 ***

The data were statistically presented as means ± SEM. The # *p* < 0.001 between CONT and LPS groups, and * *p* < 0.05, ** *p* < 0.01, *** *p* < 0.001, **** *p* < 0.0001 between LPS and ICA therapy groups. (CONT, LPS, ROS, MDA, SOD, CAT, Gpx1, and ICA stand for control, lipopolysaccharide, Superoxide dismutase, Catalase, Glutathione peroxidase 1, and Icariin groups [The ICA dose used was 50 mg/kg for uterine tissue]).

**Table 2 ijms-23-10219-t002:** Treatment design.

Treatments	Hours				
	0	24	30	36	42
CONT	PBS	_	_	_	Euthanized
LPS	LPS	_	_	_	Euthanized
LPS + ICA	LPS	ICA	ICA	ICA	Euthanized
ICA	PBS	ICA	ICA	ICA	Euthanized

(CONT, LPS, and ICA stand for control, lipopolysaccharide, and Icariin).

**Table 3 ijms-23-10219-t003:** Sequences of primers used in RT-qPCR.

Target Gene	Primer	Primer Sequence (5’ 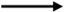 3’)	Accession No.	Product Size
*TLR4*	Forward	ATTCAGAGCCGTTGGTGTATC	NM_021297.2	109
Reverse	GGGACTTCTCAACCTTCTCAAG
*TNF-α*	Forward	GGGCTTTACCTCATCTACTCA	NM_013693.3	198
Reverse	GCTCTTGATGGCAGACAGG
*IL-1β*	Forward	CCTGGGCTGTCCTGATGAGAG	NM_008361.4	131
Reverse	TCCACGGGAAAGACACAGGTA
*IL-6*	Forward	GGCGGATCGGATGTTGTGAT	NM_031168.1	199
Reverse	GGACCCCAGACAATCGGTTG
*IL-10*	Forward	ACAGCCGGGAAGACAATAACT	NM_010548.2	66
Reverse	GCAGCTCTAGGAGCATGTGG
*SOD1*	Forward	GGTCTCCAACATGCCTCTCT	NM_011434.2	203
Reverse	AACCATCCACTTCGAGCAGA
*CAT*	Forward	CACTGACGAGATGGCACACT	NM_009804.2	175
Reverse	TGTGGAGAATCGAACGGCAA
*Gpx1*	Forward	GTACTTGGGGTCGGTCATGA	NM_001329527.1	222
Reverse	GGTTTCCCGTGCAATCAGTT
*Nrf2*	Forward	TCCTATGCGTGAATCCCAAT	NM_010902.3	103
Reverse	GCGGCTTGAATGTTTGTCTT
*HO-1*	Forward	GGGCTGTGAACTCTGTCCAATGT	NM_010442.2	162
Reverse	TTGGTGAGGGAACTGTGTCAGG
*NQO1*	Forward	TTCTGTGGCTTCCAGGTCTTAG	NM_008706.5	156
Reverse	GTCAAACAGGCTGCTTGGAGCAA
*GCLC*	Forward	ACAAGGACGTGCTCAAGTGG	NM_010295.2	199
Reverse	CCAGGCGTTCCTTCGATCAT
*GAPDH*	Forward	CAATGTGTCCGTCGTGGATCT	NM_001289726.1	124
Reverse	GTCCTCAGTGTAGCCCAAGATG

## Data Availability

Data is contained within the article.

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
