# Peer review of "Icariin Alleviates Escherichia coli Lipopolysaccharide-Mediated Endometritis in Mice by Inhibiting Inflammation and Oxidative Stress"

_ijms, 2022, doi:10.3390/ijms231810219_

Round 1
Reviewer 1 Report
This work is of interest to dairy producers, scientist and veterinarians.
The manuscript can be accepted for publication after minor revisions.
Suggested edits are shown in the attached PDF in bold and sticky notes.

Author Response
We also thank the critical comments and totally agree with your suggestions, which are of great help to improve the quality of our manuscript. We have carefully taken the comments into consideration in preparing our revision, which has made our manuscript clearer, more compelling, and broader. Here are our responses to each comment correspondingly.
- This work is of interest to dairy producers, scientist and veterinarians. The manuscript can be accepted for publication after minor revisions. Suggested edits are shown in the attached PDF in bold and sticky notes.
Response: Thank you for your extensive reading and for helping us to improve our manuscript (MS). However, all the suggested edits in bold and sticky notes have been added in the revised MS in tract changes in the Microsoft word file.
Reviewer 2 Report
General comments
In this manuscript, the authors reported their study on the effect of icariin on LPS-induced endometritis using a mouse model. The study was carefully designed, the methods were sound and results supported their conclusion. This is an interesting study and provides useful information for treatment and prevention of endometritis using the alternative therapy.
Special comments
1. The abstract. Abstract should contain a brief description of materials and methods. Suggest re-organising the abstract.
2. Lines 25-28. Rephrase.
3. Line 34. Change “messenger (m) RNA” to “mRNA” as this is a well-established abbreviation.
4. Line 38. Change “in endometritis” to “challenge”.
5. Line 41. Change “research” to “study”.
6. Line 51. Change “Additionally, misuse” to “Misuse”.
7. Lines 56-57. Change “Thus, we also utilized a mice model to study endometritis” to “In the present study, we utilized a mouse model to investigate endometritis”.
8. Line 69. Change “deliberated” to “believed”.
9. Line 70. Change “studied” to “shown”.
10. Line 72. Change “implicated” to “involved”.
11. Line 87. Change “research” to “study”.
12. Line 97. Change “that” to “and”.
13. Lines 104-105. What did it do to the W/D ratio (increased or decreased)?
14. Line 105. Change “markedly” to “significantly”.
15. Line 109. Check to see if you need to change “the MPO” to “the LPS-induced MPO”.
16. Line 111. Change “finding has” to “results have”
17. Lines 111-113. Did you mean "LPS group" vs "LPS+ICA group"? If so, you may say "the LPS-stimulated NO production were significantly down-regulated by the treatment of ICA (P<0.05)." or "the stimulatory effect of LPS on NO production was significantly inhibited by ICA treatment (P<0.05)."
18. Figures 2, 3, 4, 5, 6. What is GC in the figures? The statistical annotations are difficult to understand. Please improve it.
19. Figures 2, 3, 4, 5 and 6. Change “The #p < 0.001 is between CG vs. LPS groups, whereas *p < 0.05, **p < 0.01 LPS vs. ICA therapy groups.” to “# p < 0.001 between CG and LPS groups, and *p < 0.05, **p < 0.01 between LPS and ICA therapy groups.”
20. I suggest that you change “GC” to “Control” or “CONT” throughout the manuscript.
21. Line 119. What is “ICA therapy group”? is it “LPS+ICA” group. Please make it clear throughout the manuscript.
22. Line 128. Change “revealed” to “showed” or “demonstrated”.
23. Line 131. Delete “as”.
24. Line 131. Change “as well as” to “and”.
25. Line 140. Change “Consequence” to “Effect”.
26. Line 142. Change “finding” to “results”.
27. Line 171. Change “finding” to “results”.
28. Lines 201-203. Rephrase.
29. Line 213. Please check to see whether it is “activation” or “inhibition” of NF-kB.
30. Lines 225-230. Rephrase.
31. Lines 235-236. Change “supposed” to “considered”.
32. Line 241. Change “recover” to “improve”.
33. Lines 243-244. Please check to see if this statement is appropriate as most of the anti-inflammatory drugs may not affect IL10 expression.
34. Line 247. Change “results of” to “previous reports”.
35. Line 248. Change “in” to “of”.
36. Line 271. Delete “very”.
37. Line 274. Delete “by Ref”.
38. Line 300. Change “observed LPS” to “observed that LPS”.
39. Line 302. Change “earlier research” to “previous study”.
40. Line 313. Change “obtained” to “obtained from”.
41. Line 332. Change “designated” to “described”.
42. Lines 363-371. The sources of reagents/kits (such as manufacturer/supplier, address) should be given.
43. Line 368. The protocol of qRT-PCR should be described.
44. Line 368. The primer design should be given.
45. Line 370. “-DDCt” should be superscript.
46. Lines 389-393. Homogeneity of variance should be tested before parametric tests (e.g. ANOVA). If the data between groups are not homogeneous, non-parametric tests should be used.
47. Lines 395-399. Rephrase.
Author Response
We thank the critical comments and totally agree with your suggestions, which are of great help to improve the quality of our manuscript. We have carefully taken the comments into consideration in preparing our revision, which has made our manuscript clearer, more compelling, and broader. Here are our responses to each comment correspondingly.
In this manuscript, the authors reported their study on the effect of icariin on LPS-induced endometritis using a mouse model. The study was carefully designed, the methods were sound and results supported their conclusion. This is an interesting study and provides useful information for treatment and prevention of endometritis using the alternative therapy.
Response: We are highly thankful to worthy reviewer for critically reviewing our article and helping us to improve our MS quality. However, here is a point-by-point response marked in blue color.
Special comments
- The abstract. Abstract should contain a brief description of materials and methods. Suggest re-organising the abstract.
Response: Thanks for the reviewer's comment. Abstract has been reorganized in the revised MS, containing the description of material and methods (please see the lines 30-35).
- Lines 25-28. Rephrase.
Response: We have modified the paragraph in the revised MS (please see the lines 25-28).
- Line 34. Change “messenger (m) RNA” to “mRNA” as this is a well-established abbreviation.
Response: Thanks for a nice suggestion. It has been changed to “mRNA”on line 41.
- Line 38. Change “in endometritis” to “challenge”.
Response: Thanks for a nice comment. This sentence has been changed to “challenge”on line 46.
- Line 41. Change “research” to “study”.
Response: It has been changed to “study” on line 48.
- Line 51. Change “Additionally, misuse” to “Misuse”.
Response: We have incorporated your suggestion in the revised MS on line 59.
- Lines 56-57. Change “Thus, we also utilized a mice model to study endometritis” to “In the present study, we utilized a mouse model to investigate endometritis”.
Response: Thanks for your helpful comment. It has been changed to “In the present study, we utilized a mouse model to investigate endometritis” on lines 64-65.
- Line 69. Change “deliberated” to “believed”.
Response: The suggestion have been modified “deliberated” to “believed” on line 77.
- Line 70. Change “studied” to “shown”.
Response: It has been changed to “Shown” on line 78.
- Line 72. Change “implicated” to “involved”.
Response: It has been changed to “involved” on line 81.
- Line 87. Change “research” to “study”.
Response: It has been changed to “study” on line 95.
- Line 97. Change “that” to “and”.
Response: It has been changed to “and” on line 107.
- Lines 104-105. What did it do to the W/D ratio (increased or decreased)?
Response: Thanks for your helpful comment. LPS increased the W/D ratio; its also been mention in revised MS on line 115.
- Line 105. Change “markedly” to “significantly”.
Response: It has been changed to “significantly” on line 115.
- Line 109. Check to see if you need to change “the MPO” to “the LPS-induced MPO”.
Response: Thank you for a helpful comment. According to your seggection we have added “the LPS-induced MPO” in the revised MS on line 119.
- Line 111. Change “finding has” to “results have”
Response: It has been changed to “results have” on line 122.
- Lines 111-113. Did you mean "LPS group" vs "LPS+ICA group"? If so, you may say "the LPS-stimulated NO production were significantly down-regulated by the treatment of ICA (P<0.05)." or "the stimulatory effect of LPS on NO production was significantly inhibited by ICA treatment (P<0.05)."
Response: Thank you for the helpful suggestion. Yes, we means "LPS group" vs "LPS+ICA as well as ICA group”. According to your suggestion, we have addedin the revised MS “the LPS-stimulated NO production was significantly down-regulated by the treatment of ICA (P<0.05)” on lines 123-124.
- Figures 2, 3, 4, 5, 6. What is GC in the figures? The statistical annotations are difficult to understand. Please improve it.
Response: Thank you for a helpful suggestion. According to your question/ suggestion (at comment no. 20), we have changed “GC” to “CONT” throughout the revised MS. Modified figures have been reinserted in the revised MS for the said revision.
- Figures 2, 3, 4, 5 and 6. Change “The #p < 0.001 is between CG vs. LPS groups, whereas *p < 0.05, **p < 0.01 LPS vs. ICA therapy groups.” to “# p < 0.001 between CG and LPS groups, and *p < 0.05, **p < 0.01 between LPS and ICA therapy groups.”
Response: Thank you for the nice comment. We have changed it to “# p < 0.001 between CG and LPS groups, and *p < 0.05, **p < 0.01 between LPS and ICA therapy groups.” In the Figures 2, 3, 4, 5 and 6.
- I suggest that you change “GC” to “Control” or “CONT” throughout the manuscript.
Response: Thank you for a helpful suggestion. We have changed “GC” to “CONT” throughout the revised MS according to your suggestion.
- Line 119. What is “ICA therapy group”? is it “LPS+ICA” group. Please make it clear throughout the manuscript.
Response: Thank you for the helpful suggestion. ICA therapy group means those groups which received the ICA therapy in the form of injection, such as LPS+ICA and ICA groups.
- Line 128. Change “revealed” to “showed” or “demonstrated”.
Response: We have modified as suggested by worthy reviewer in the revised MS (please see the line 141.
- Line 131. Delete “as”.
Response: We have deleted (please the line 144).
- Line 131. Change “as well as” to “and
Response: It has been changed to “and” on line 144.
- Line 140. Change “Consequence” to “Effect”.
Response: It has been changed to “Effect” on line 154.
- Line 142. Change “finding” to “results”.
Response: It has been changed to “results” on line 156.
- Line 171. Change “finding” to “results”.
Response: It has been changed to “results” on line 189.
- Lines 201-203. Rephrase.
Response: It has been rephrased on lines 222-224.
- Line 213. Please check to see whether it is “activation” or “inhibition” of NF-kB.
Response: Sorry, it is our mistake, it's actually “inhibition”, we have modified the paragraph in the revised MS (Please see the line 238).
- Lines 225-230. Rephrase.
Response: Thanks for the reviewer's comment. These sentences have been rephrased in the revised MS (please see the lines 250-260).
- Lines 235-236. Change “supposed” to “considered”.
Response: It has been changed to “considered” on line 266.
- Line 241. Change “recover” to “improve”.
Response: It has been changed to “improve” on line 271.
- Lines 243-244. Please check to see if this statement is appropriate as most of the anti-inflammatory drugs may not affect IL10 expression.
Response: Thanks for the reviewer's comment. We agree with your statement and we have modified our sentence in the revised MS (please see lines 274-275).
- Line 247. Change “results of” to “previous reports”.
Response: It has been changed to “previous reports” on line 278-279.
- Line 248. Change “in” to “of”.
Response: It has been changed to “of” on line 280.
- Line 271. Delete “very”.
Response: It has been deleted from line 303.
- Line 274. Delete “by Ref”.
Response: It has been deleted from line 306.
- Line 300. Change “observed LPS” to “observed that LPS”.
Response: It has been changed to “observed that LPS” in the revised MS (please see the line 332).
- Line 302. Change “earlier research” to “previous study”.
Response: It has been changed to “previous studies” on line 334.
- Line 313. Change “obtained” to “obtained from”.
Response: It has been changed to “obtained from” on line 345.
- Line 332. Change “designated” to “described”.
Response: It has been changed to “described” on line 365.
- Lines 363-371. The sources of reagents/kits (such as manufacturer/supplier, address) should be given.
Response: The sources of reagents/kits (such as manufacturer/supplier, address) have been mentioned on lines 398-410.
- Line 368. The protocol of qRT-PCR should be described.
Response: The detailed protocol of qRT-PCR has been described on lines 398-410.
- Line 368. The primer design should be given.
Response: The primer design has been mentioned in table 3 (line 411).
- Line 370. “-DDCt” should be superscript.
Response: It has been superscripted on line 410.
- Lines 389-393. Homogeneity of variance should be tested before parametric tests (e.g. ANOVA). If the data between groups are not homogeneous, non-parametric tests should be used.
Response: Thanks for the reviewer's comment. All the data was subjected to testing homogeneity and parallelism for statistical analysis. After analysis, it was determined that data fullfil all the requirements be subjected to One-way ANOVA.
- Lines 395-399. Rephrase.
Response: Thanks for the reviewer's comment. These sentences have been rephrased in the revised MS (lines 435-442).
Reviewer 3 Report
In the title you should add the species (mice model).
Introduction - in many places it is not known what species it is, so the information should also be more precise.
Figure 2 The #p <0.001 is between CG vs. LPS groups, whereas * p <0.05, ** p <0.01. Generally p <0.001 is marked as ***. There are *** in the figure as well and they are not explained, so they should also add figures in the description.
There are **** in table 1 not explained?
L200 to L211 is more of an introduction, not a discussion.
The paragraph "animals" must be added. Please describe in detail the procedures that were performed on the animals.
Author Response
We thank the critical comments and totally agree with your suggestions, which are of great help to improve the quality of our manuscript. We have carefully taken the comments into consideration in preparing our revision, which has made our manuscript clearer, more compelling, and broader. Here are our responses to each comment correspondingly.
- In the title you should add the species (mice model).
Response: Thanks for a nice comment. The species name in the title has been added (please see the lines 2-3).
- Introduction - in many places it is not known what species it is, so the information should also be more precise.
Response: Thanks for the worthy reviewer comment. Species names have been added in the revised manuscript (please see the lines 78, 79, 80 and 82).
- Figure 2 The #p <0.001 is between CG vs. LPS groups, whereas * p <0.05, ** p <0.01. Generally p <0.001 is marked as ***. There are *** in the figure as well and they are not explained, so they should also add figures in the description.
Response: Thank you for your valuable comment. ***p have been explained in the description of figure 2 (please see the lines 132-133).
- There are **** in table 1 not explained?
Response: The symbols *** has been explained in the revised MS on description table 1 (please see the line 196).
- L200 to L211 is more of an introduction, not a discussion.
Response: We are very much thankful for such constructive comments on the manuscript. We have modified and summarized the first paragraph of the discussion section from L200 to L211 into 2-3 sentences. We supposed 2-3 sentences are necessary as an opening statement at the start of the discussion section (on lines 221-227).
- The paragraph "animals" must be added. Please describe in detail the procedures that were performed on the animals.
Response: Thanks for the constructive comment about improving the quality of our MS. A separate heading, "animals" have been added in the revised MS that describes all the procedures performed on the animals in detail (please see the lines 354-359).